# Integrating Proteomics and Transcriptomics Reveals the Potential Pathways of Hippocampal Neuron Apoptosis in Dravet Syndrome Model Mice

**DOI:** 10.3390/ijms25084457

**Published:** 2024-04-18

**Authors:** Xuerui Kong, Gaohe Dai, Zhong Zeng, Yi Zhang, Jiarong Gu, Teng Ma, Nina Wang, Jinhai Gu, Yin Wang

**Affiliations:** 1School of Basic Medicine, Ningxia Medical University, Yinchuan 750004, China; kxuerui1115@163.com (X.K.); maten@nxmu.edu.cn (T.M.); nina1823c@163.com (N.W.); 2Key Laboratory of Craniocerebral Diseases, Ningxia Medical University, Yinchuan 750004, China; daigaohe2021@163.com (G.D.); zhangyiiii9@163.com (Y.Z.); 3School of Clinical Medicine, Ningxia Medical University, Yinchuan 750004, China; zlsdys1995@163.com; 4School of Public Health, Ningxia Medical University, Yinchuan 750004, China; gujiarong2023@163.com

**Keywords:** Dravet syndrome, *Scn1a* mice, proteomics, transcriptomics, neuron apoptosis, VHL, HIF-1α, p21

## Abstract

An important component contributing to the onset of epilepsy is the death of hippocampal neurons. Several studies have shown that Dravet syndrome model mice: *Scn1a* KO mice have a high number of apoptotic neurons following seizures, but the precise mechanism underlying this remains unclear. The aim of this research was to elucidate the potential molecular mechanism of neuronal apoptosis in *Scn1a* KO mice by integrating proteomics and transcriptomics, with the ultimate goal of offering better neuroprotection. We found that apoptotic processes were enriched in both proteomic and transcriptomic GO analyses, and KEGG results also indicated that differential proteins and genes play a role in neurotransmission, the cell cycle, apoptosis, and neuroinflammation. Then, we examined the upstream and downstream KGML interactions of the pathways to determine the relationship between the two omics, and we found that the HIF-1 signaling pathway plays a significant role in the onset and apoptosis of epilepsy. Meanwhile, the expression of the apoptosis-related protein VHL decreased in this pathway, and the expression of p21 was upregulated. Therefore, this study suggests that VHL/HIF-1α/p21 might be involved in the apoptosis of hippocampal neurons in *Scn1a* KO mice.

## 1. Introduction

Epilepsy is a neurological disease that impacts over seventy million individuals globally [1]. There are various factors that can cause epilepsy. The International League Against Epilepsy redefined six primary categories of epilepsy causes, namely structural, genetic, metabolic, immune, infectious, and unexplained causes, with an emphasis on the genetic origins of epilepsy. In total, 40–50% of epilepsy cases remain unexplained, with around 30% being linked to genetic factors [2]. The type of genetic epilepsy most commonly investigated is Dravet syndrome (DS) [3]. Dravet syndrome, previously called severe myoclonic epilepsy of infancy (SMEI), was initially described by Dr. Dravet in 1978. Because not all patients exhibit a myoclonic phenotype, it was later renamed Dravet syndrome by the International League against Epilepsy [4,5]. Dravet syndrome is considered one of the most severe forms of epileptic encephalopathy, with an estimated incidence rate ranging from 1/15700 to 1 /14000 [6]. Approximately 80% of cases are attributed to a mutation in the sodium channel type 1 (*SCN1A)* gene. The predominant and distinctive symptom of Dravet syndrome is heat-induced seizures, which typically start in nearly every affected child within their first year of life. These seizures are commonly induced by fevers, vaccinations, and hot baths [7,8], and then they gradually develop into long-term refractory generalized tonic–clonic seizures. This is often accompanied by severe cognitive impairment and behavioral deficits [9,10]. DS patients have a high mortality rate, which is 5.1 times higher than that of adults with refractory epilepsy [11]. Patients with DS are usually treated with medications, but traditional antiepileptic drugs, such as carbamazepine and lamotrigine (sodium-channel blockers), can exacerbate seizures and even lead to the further deterioration of cognitive deficits. The emergence of new antiepileptic drugs (cannabidiol, fenfluramine, and stavudine) seems to be a means of giving patients more choice, but 30% of patients still have unsatisfactory seizure control [12,13,14,15]. Therefore, the exploration of individualized treatment measures for patients with DS remains an urgent issue needing to be addressed. The most recent research on the impacts of antiepileptic medications has concentrated on neurons, ion channels, and transporters, along with excitatory and inhibitory neurotransmission. However, these studies only modify immediate seizure activity and do not modify the underlying spontaneous recurrent seizures. Furthermore, they do not address the neurodegenerative processes triggered by status epilepticus [16].

Mice carrying heterozygous loss-of-function mutations in *SCN1A* (*Scn1a*^+/−^) have exhibited numerous traits associated with DS, such as seizures induced by hyperthermia, spontaneous seizures, cognitive and behavioral impairments, and premature mortality [17,18]. One study found that *Scn1a* knockout (KO) mice showed significant neuronal loss in cortical regions, and treatment with medicine mitigated neuronal damage and further improved their seizures and cognitive behavioral deficits [19]. This suggests that neuronal apoptosis plays a vital role throughout the process of epileptogenesis development in *Scn1a* KO mice. In patients with epilepsy, during convulsive seizures, the hippocampus undergoes ischemia and hypoxia, resulting in injury to the hippocampal neurons and potential apoptosis [20]. In both human patients and animal models, hippocampal neuronal loss is frequently the result of epileptic episodes. In epileptic patients, the epileptiform discharges of neuronal cells can speed up the process of apoptosis in the hippocampal region, potentially due to the abnormal expression of multiple sets of apoptosis-related proteins in response to prolonged seizures, which leads to the misregulation of relevant signaling pathways [21]. WangL et al. discovered that, in PTZ-induced epilepsy model mice, there was damage to the hippocampal neurons and an increase in the expression of proteins linked to apoptosis [22]. Li Qian et al. found that using KA-induced epilepsy animal models resulted in hippocampal neuron apoptosis and glial cell death, ultimately leading to brain damage [23]. It is clear that seizures are strongly linked to hippocampal neuronal apoptosis, although the precise mechanism of seizures remains uncertain. Thus, elucidating the possible molecular mechanisms responsible for hippocampal neuronal apoptosis following seizures could facilitate the development of more effective neuroprotective strategies.

The fields of pathogenic mechanism investigation and prospective therapeutic target identification have found a new avenue with the advent of transcriptomics and proteomics in recent times [24,25]. Several molecular markers associated with epilepsy have been discovered, and the mechanism of epilepsy has been investigated using single-omics methods [26,27,28]. The progression of this disease is not attributable to a singular factor; rather, it is a complex and ongoing process that encompasses multiple levels of data alteration. It arises from a concatenation of elements, including genetics, mRNA, proteins, and metabolites. The onset and progression of epilepsy constitute a dynamic and intricate phenomenon. Using any single level of data makes it difficult to fully reveal the course of disease development. By conducting transcriptomic analyses, we can acquire pertinent data on mRNA levels in gene transcription and delve into the correlation between gene expression and pathology. Proteomic analyses allow for the examination of the shifts in protein composition and expression levels within cells and tissues at the gene translation level, thus enabling a deeper understanding of protein interactions. The process of gene expression entails more than just a linear progression from the transcriptome to the proteome; rather, it involves a complex interplay between the two. The examination of gene expression coherence and the discovery of the post-transcriptional regulation of gene expression can be aided by the integration of both transcriptomic and proteomic expression data.

In this research, we utilized proteomic and transcriptomic methodologies to investigate alterations in protein and transcription levels following heat-induced seizures in scn1a gene-knockout mice. Additionally, in order to identify putative targets and pathways connected to the death of hippocampal neurons in scn1a gene-knockout mice, we carried out an integrated analysis of transcriptomes and proteins. This offers a novel approach for the treatment or alleviation of Dravet syndrome.

## 2. Results

### 2.1. Genotype Verification of Scn1a KO Mice

We obtained F1 heterozygote mice with a C57BL6J background, resulting from the hybridization of two mouse strains, that exhibited typical features of Dravet syndrome. Subsequently, genotyping of the F1-generation mice was performed, and primers were created and manufactured using the *Scn1a* mouse gene’s exon 1 sequence as a guide. This involved the design of wild-type reverse primers, mutant reverse primers, and normal forward primers. The PCR amplification results revealed a range of 200 bp–357 bp for the PCR products of the heterozygous mice, while this figure for the wild-type (WT) mice was 357 bp (Figure 1).

### 2.2. Seizures in the Mice and EEG Recordings

We assessed the epileptic behavior of the *Scn1a* KO mice and WT mice using the Racine scoring criteria and electroencephalography during hyperthermia-induced seizures. Our findings revealed that, as the temperature increased, the WT mice exhibited anxious behaviors, such as constant jumping, with the EEG frequency and amplitude remaining at baseline levels. In contrast, 9 out of the 10 Scn1a KO mice displayed seizures of grade 4 or higher, accompanied by a significantly higher EEG frequency and amplitude than the WT mice (Figure 2; Appendix A).

### 2.3. Histologic Changes after Seizures in Scn1a KO Mice

Using Nissl and TUNEL labeling, we evaluated the neuronal changes in the hippocampal tissues of the *Scn1a* KO mice and WT mice in order to detect apoptosis in their hippocampi. The Nissl staining results revealed that the neurons in the hippocampal ca3 and ca1 regions of the *Scn1a* KO mice were sparsely distributed and decreased in number, displaying cellular condensation, nuclear fragmentation, and lysis, compared to those of the WT mice (*p* < 0.05, Figure 3a–c). In the TUNEL staining, cells with positive staining (green fluorescence) were observed in the ca3 and ca1 regions of the *Scn1a* KO mice, while positive cells were nearly absent in the WT mice (*p* < 0.05, Figure 3d–f).

### 2.4. Proteomics Results

Based on proteomic data, the sample distribution of the mice in the KO and WT groups was represented using a principal component analysis (PCA) (Figure 4a). A total of 7155 proteins were analyzed in the *Scn1a* KO mice and WT mice. Among these, 6630 were considered plausible proteins (Appendix A). Based on a fold change ≥1.2 or a fold change ≤1/1.2 and *p* < 0.05 for the significance of the different standards, the screening of the KO and WT groups revealed differences between the proteins. A total of 37 differentially expressed proteins (DEPs) were identified, among which 16 were upregulated and 21 were downregulated (Appendix A). The blue dot on the left of the differential expression volcano map represents the significantly downregulated genes, and the red dot on the right represents the significantly upregulated genes (Figure 4b). In order to visualize protein expression, the DEPs of each group were examined using a hierarchical cluster analysis, and a heatmap was created, with red representing proteins that were highly expressed and blue indicating proteins that were expressed at low levels. The 37 different ID names on the right of the figure represent the 37 different proteins. The corresponding protein names can be seen in Appendix A (Figure 4c).

In order to clarify how the differentially expressed proteins in the pathophysiological processes of epileptogenesis and post seizure in the Scn1a KO mice are regulated physiologically, and to acquire a thorough understanding of these proteins’ roles, locations, and biological processes, GO functional enrichment analyses were conducted. The top 10 GO enrichment functions for both downregulated and upregulated proteins are shown in bar charts in Figure 4d,e. The downregulated DEPs were enriched in the apoptotic process, long−term synaptic potentiation regulation, and protein ubiquitination in the biological processes. Meanwhile, the upregulated DEPs were enriched in the extrinsic apoptotic signaling pathway, the negative regulation of the apoptotic process, and the positive regulation of cell population proliferation. The KO mice appeared to exhibit more distinct functional variations, particularly in apoptosis, than the WT mice. To obtain a more thorough and organized understanding of the biological mechanisms and related processes of the *Scn1a* KO mice, we conducted an analysis and annotation of all of the differential proteins using the KEGG database. Regarding the downregulated proteins, the DEPs mainly involved the dopaminergic synapse, ubiquitin-mediated proteolysis, the cell cycle, and the HIF-1 signaling pathway (Figure 4f). Regarding the upregulated proteins, the DEPs mainly involved neuroactive ligand–receptor interactions and the complement and coagulation cascades (Figure 4g). The KEGG results reveal that DEPs play a role in synaptic function, neurotransmission, the cell cycle, and immune function.

Four randomly selected differential proteins were subjected to a WB experiment in order to verify the accuracy of the proteomic data. The results of the WB tests showed that the differential protein trends were in line with the proteomic data. Specifically, Ace, Aqp1, and Folr1 were found to be upregulated. In comparison to the mice in the WT group, VHL was downregulated in the hippocampus tissue of the KO group, confirming the validity of the proteomic data (*p* < 0.05, Figure 5a–d).

### 2.5. Transcriptomic Results

A total of 19903 genes were analyzed in the *Scn1a* KO mice and WT mice (Appendix A). The transcriptomic data revealed that the principal component analysis (PCA) (Figure 6a) accurately represented the sample distribution of the mice in the KO group and WT group. Closer proximity and aggregation within the groups indicated smaller differences and greater similarities, while greater distances reflected larger differences between the samples. The figure clearly demonstrates improved sample aggregation and notable differences between the KO mice and WT mice. Based on a fold change ≥2 or a fold change ≤0.5 and *q* < 0.05 for the significance of the different standards, the screening of the KO and WT groups revealed differences between the genes. A total of 733 differentially expressed genes (DEGs) were screened, among which 524 were upregulated and 209 were downregulated (Figure 6b and Appendix A). The blue dot located on the left side of the volcano map for differential expression represents the top 10 genes that were significantly downregulated, while the red dot on the right side indicates the top 10 genes that were significantly upregulated (Figure 6b). The protein expression level was visualized using a hierarchical cluster analysis heatmap created from the DEGs of each group, with orange/red representing the genes that were highly expressed and blue indicating the genes that were expressed at low levels. Among the 20 genes on the right side of the figure, the first 10 were upregulated genes and the last 10 were downregulated genes (Figure 6c).

We conducted a GO enrichment analysis of the differential genes to investigate the physiological regulation of the genes involved in the pathogenesis and pathophysiology of the *Scn1a* KO mice. The top 10 GO enrichment functions for both the upregulated and downregulated genes are shown using bar diagrams. In terms of biological processes, the downregulated DEGs were enriched in the neuropeptide signaling pathway, the negative regulation of glial cell apoptosis, and the negative regulation of neurogenesis. Concurrently, the upregulated DEGs were enriched in the extracellular matrix architecture, the negative regulation of the apoptotic process, the cytokine response, and other related processes (Figure 6d,e). In order to learn more about how the differentially expressed genes in the *Scn1a* KO mice function biologically, we utilized the KEGG database to conduct a pathway analysis on the entire set of differentially expressed genes. Regarding the downregulated genes, the DEGs mainly involved neuroactive ligand–receptor interactions, the TGF-beta signaling pathway, the Notch signaling pathway, and the Wnt signaling pathway (Figure 6f). Regarding the upregulated genes, the DEGs mainly involved the MAPK signaling pathway, PI3K-Akt signaling pathway, TNF signaling pathway, NF-kappa B signaling pathway, P53 signaling pathway, IL-17 signaling pathway, and neuroactive ligand–receptor interactions (Figure 6g). The KEGG results indicate that DEGs are involved in neurotransmission, the cell cycle, apoptosis, and neuroinflammation. Chord diagrams were used to display the enrichment of the 10 categories with the lowest *q*-values in the KEGG pathway analysis, as demonstrated in the figure. The left side illustrates the 10 most significant genes in logFC for each category, the right side represents the composition of the classifications, and the middle line signifies the correspondence between the classifications and genes (Appendix A).

Five genes were randomly selected to verify the accuracy of the transcriptomic data obtained using RT-qPCR. The RT-qPCR results showed that the genes exhibited trends consistent with those of the transcriptomic data. Specifically, the results revealed that, compared to the WT mice, the expressions of Etv3l, Sprrla, Tgm1, and Trh were upregulated, while the expression of Cartpt was downregulated in the hippocampi of the KO mice, confirming the validity of the transcriptomic data (Figure 7a–e).

### 2.6. Integrative Analysis of Proteomics and Transcriptomics

We conducted a correlation study between the transcriptomic and proteomic data. Initially, we categorized genes and proteins as connected pairs in terms of their relationships. The corresponding method was as follows: When a gene and protein came from the same reference genome or the protein sequence was translated from the transcript, the gene and protein were associated according to the annotation or ID correspondence. When the gene and protein were not directly related, the association was based on the gene name (protein name) or blast sequence homologous alignment. Venn diagrams were employed to examine the overlap of shared genes, differentially expressed genes, and differentially expressed proteins that were both upregulated and downregulated. Figure 8a indicates that only two of the upregulated omics overlap, with no overlap observed in the downregulated omics. The connection between the proteomes and transcriptomes was found to be limited, aligning with findings from past research on discrepancies between the two. The biological effects of DEP enrichment involved cell division, cell death, and synaptic activity, while DEG enrichment primarily involved signaling pathways, apoptosis regulation, and the cytokine response. The biological roles of the DEPs and DEGs both highlight cell apoptosis, leading us to investigate their shared pathways. All DEGs and DEPs were mapped to the Kyoto Encyclopedia of Genes and Genomes (KEGG) pathway database, and information on their common pathways was acquired. A total of 32 co-expression pathways were obtained (Table 1).

Then, we examined the upstream/downstream interactions with KGML. The KGML (modified Extensible Markup Language or XML format) file is a subset of the KEGG database, and it contains both the relationships of graphic objects in the KEGG pathway database and the information of orthologous genes in the KEGG GENES database. These details can be utilized to reveal the connections between the pathways housing the differential genes of the two omics, and these connections can be depicted in a network to facilitate a more systematic exploration of the interactions between the two omics and to provide a deeper understanding of the signaling pathways that simultaneously change across multiple omics. A statistical analysis was performed on the interconnected components of the pathways, and the top 30 pathways were identified based on the number of interconnected components. The top five pathways consisted of the cell cycle, complement and coagulation cascades, the HIF-1 signaling pathway, the ubiquitin-mediated proteolysis pathway, and coronavirus disease—COVID-19 (Figure 8b). An interaction network diagram was drawn for the interaction results of the KEGG signal paths. The node name is the path name, the node size indicates the connection degree, and the node color is identified by the KEGG Class of the path, where one is level 1 (Figure 8c) and the other is level 2 (Figure 8d). We found that the HIF-1 signaling pathway plays a vital role in the onset and apoptosis of epilepsy (see the Discussion section for details). In the HIF-1 signaling pathway identified in the KEGG pathway database (mmu04066) (Figure 9), the expression of vhl was decreased, while the expression of CDKN1A(p21) was increased. We confirmed the involvement of these proteins and genes in these pathways. The Western blot results revealed that VHL expression decreased (Figure 5c), HIF-1α expression increased, and p21 expression increased in the hippocampi of the KO mice. Additionally, our RT-qPCR results also demonstrated an increase in the p21 gene’s expression (Figure 10a–d).

## 3. Discussion

Currently, the pathophysiology of epilepsy remains elusive. Numerous studies have shown that a considerable amount of neuronal apoptosis occurs following epilepsy, with hippocampal neuron apoptosis being a significant form of neuronal loss after seizures [29]. The hippocampus is a vital anatomical component for short-term memory within the central nervous system, as well as serving as the foundation for memory and learning [30]. The hippocampus is highly susceptible to injury, and the initial manifestation of epilepsy frequently originates in the hippocampus. Therefore, it is essential to investigate the regulatory mechanisms behind and the link between hippocampal neuronal death and the onset of epilepsy. In this research, we used *Scn1a* KO mice, which served as a mouse model for investigating hereditary epilepsy, specifically Dravet syndrome. Compared to WT mice from the same litter, we observed cellular shrinkage, nuclear fragmentation, and dissolution in the hippocampus of the Scn1a KO mice in Nissl staining. The quantity of Nissl staining-positive cells in the CA1 and CA3 regions of the hippocampus was reduced in the KO mice compared to in the WT mice. Tunnel staining revealed extensive green fluorescence in both the CA3 and CA1 regions of the hippocampus in the *Scn1a* KO mice, suggesting neuronal apoptosis. These findings are in line with those of Liu’s study [19], and they indicate that hippocampal neuronal apoptosis occurs in *Scn1a* KO mice following heat-induced seizures.

With the development of high-throughput technologies, researchers have been able to use a variety of genomic techniques to better understand how diseases evolve [31]. Therefore, in our study, we integrated proteomics and transcriptomics to perform a unique analysis of hippocampal tissues obtained from *Scn1a* KO mice and their littermate WT mice in order to further investigate the apoptotic process in hippocampal neurons. This is the first time such an analysis has been conducted. Initially, we used proteomic and transcriptomic techniques to identify differential proteins and differential genes after heat-induced seizures in the *Scn1a* KO mice and WT mice, and we performed GO and KEGG enrichment analyses on them, respectively. The GO results showed that, in terms of biological processes, in the bar diagram of the downregulated DEP enrichment analysis (Figure 4d), the top 10 DEPs were included in apoptosis. In the bar diagram of the upregulated DEP enrichment analysis (Figure 4e), the top 10 DEPs were included in the extrinsic apoptotic signaling pathway and negative regulation of apoptosis. In addition, in the downregulated DEPs, Eya3 participated in the negative regulation of the extrinsic apoptotic signaling pathway in the absence of ligands. In the upregulated DEPs, TP63 participated in the positive regulation of the apoptotic signaling pathway. We found that both upregulated and downregulated DEPs were involved in pro-apoptotic and anti-apoptotic aspects. The balance between apoptosis and anti-apoptosis mechanisms is important for the normal growth and development of organisms, as well as for the treatment of diseases. On the other hand, the dysregulation of apoptosis occurs in a variety of disease states, including autoimmune diseases, neurodegenerative diseases, and epilepsy, and contributes to the pathology of these diseases [32,33,34]. These results suggest that DEPs may be involved in the dysregulation of apoptosis in hippocampal neurons in *Scn1a* mice.

In terms of biological processes, in the bar diagram of the downregulated DEG enrichment analysis (Figure 6d), the top 10 DEGs were involved in the negative regulation of glial cell apoptosis. In the bar diagram of the upregulated DEG enrichment analysis (Figure 6e), the top 10 DEGs were involved in the negative regulation of apoptosis. It has been reported that seizure-induced neuronal death is often accompanied by the activation of microglia and astrocytes [35,36]. Under physiological conditions, activated microglia usually have a phagocytic function in the central nervous system, which can engulf apoptotic and necrotic neurons and foreign viruses and bacteria [37]. However, under pathological conditions, the number of activated microglia increases, and pro-inflammatory factors such as TNF-α and il-6 are produced; the increase in these inflammatory factors is the key to the apoptosis and necrosis of neurons [38]. After neuronal injury, astrocytes show a pathological increase, and the expression of their specific marker, GFAP, also increases [39]. Astrocytes are the main components of epileptic mechanisms. In the early stage after an epileptic seizure, astrocytes may play a role in protecting neurons, but, in the late stage, astrocytes also release a large number of inflammatory factors, accelerating the death of neurons [40]. These results suggest that DEGs may be involved in the apoptosis of hippocampal neurons in *Scn1a* mice. The KEGG results show that the DEPs play a role in synaptic function, neurotransmission, the cell cycle, and immune function, and that the DEGs play a role in neurotransmission, the cell cycle, apoptosis, and neuroinflammation, which is consistent with the results of Huang et al. [41].

We performed a correlation study between the transcriptomes and proteomes and found that the association between them was limited, consistent with the results of previous studies on the differences between the two. Therefore, all DEGs and DEPs were mapped to the KEGG pathway database, and information on their common pathways was obtained. We then examined the pathway upstream/downstream interactions with KGML, and the top five pathways were found to be the cell cycle, complement and coagulation cascades, HIF-1 signaling pathway, ubiquitin-mediated proteolysis, and coronavirus disease—COVID-19.

To date, many studies have proven that the causes of neuronal apoptosis also include the abnormal cell cycle of neurons. Reactivation of the cell cycle component may cause neuronal death in the developing brain and neuronal degeneration in the adult brain, which is frequently seen in dead neurons [42,43,44]. The idea that apoptosis results from a neuron’s failed attempt to enter the cell cycle is based on reports suggesting that the cell cycle process may be strongly linked to the death of terminally differentiated neurons [45,46,47]. The coagulation and complement cascade signaling networks are essential for preserving immunological function, and the complement cascade has been reported to promote inflammation and apoptosis [48,49]. At the same time, the coagulation cascade plays an important role in neurological diseases [50]. Previous studies have demonstrated the involvement of the HIF-1 signaling pathway in the development of epilepsy [51,52], and further exploration of its apoptotic mechanisms in hippocampal neurons during seizures is needed. HIF-1 is a heterodimer, and it contains both α and β subunits and is widely present in almost all cells [53]. HIF-1 is strongly affected by oxygen levels, has a very short half-life, and is barely detectable under normoxic conditions, where HIF-1α is hydroxylated by the prolyl hydroxylase superfamily (PHD), and hydroxylated HIF1α can be further ubiquitin-degraded by VHL-VCB-CUL2 complexes. However, under hypoxic conditions (or when VHL proteins are nonfunctional), the actions regulated by VHL are suppressed, leading to the accumulation of HIF1α within the cell. Because of this accumulation, HIF-1α can bind to hypoxia-responsive elements (HREs) on the genome in a heterodimer with HIF-1β. This, in turn, triggers the transcription of downstream genes involved in pathological processes, such as cell proliferation and apoptosis [54,55]. The VHL gene encodes a functional protein consisting of two protein isoforms: VHL30 and VHL19. VHL is a crucial component of the E3 ubiquitin ligase complex, which forms the VCB-CUL2 complex by binding to Elongin B, Elongin C, and Cullin2 (CUL2), as well as Rbx1. This complex is similar to the SCF complex (Skpl-Cdc53-F-box) in yeast [56]. Maxwell et al. [57] first revealed the essential function of VHL proteins in controlling HIF1α and provided a comprehensive understanding of the VHL-HIF-1α signaling pathway and the VCB-CR complex-mediated ubiquitination of HIF1α. Another research study indicated that mice lacking the VHL gene were capable of maintaining stable levels of HIF-1 expression under normal oxygen conditions [58]. It is commonly recognized that the COVID-19 virus can reach the central nervous system by either using anticlinal transport routes or by infecting motor or sensory neurons [59]. The virus initiates a massive inflammatory cascade, activates microglia, and causes reactive astrocyte growth. Pro-inflammatory cytokines (TNF-α, IL-6, and IL-1β), nitric oxide, prostaglandin E2, and free radicals are released when the virus enters the central nervous system, resulting in chronic inflammatory hyperneuria, seizures, and death. An increase in inflammatory factors also worsens neuronal necrosis and apoptosis in the central nervous system, particularly in several hippocampal regions. These pro-inflammatory factors are crucial in the development of epilepsy [60,61]. At present, the role of COVID-19 in the occurrence of epilepsy is not clear, and a large number of studies on this are needed.

Combining the above conclusions, we found that VHL plays a crucial role in the HIF-1 signaling pathway and ubiquitin-mediated proteolysis, as well as directly linking the two pathways. We observed a downregulated expression of the differential protein vhl and an upregulated expression of the differential gene p21 in the HIF-1 signaling pathway (mmu04066), a common pathway mapped in the KEGG pathway database. p21 is a cyclin-dependent kinase suppressor protein (CDKI), and it plays a dual role in cell apoptosis [62]: the p21 protein can inhibit the anti-apoptotic effect of caspase-2 upstream of caspase-3 [63], and it can also inhibit apoptosis by inhibiting the activation of caspase-dependent CDK, blocking the chromatin concentration and cell wrinkling [64]. By activating the TNF family death receptor (TRAIL) or upregulating the pro-apoptotic protein Bax, the p21 protein can induce cell death [65]. Chen et al. [66] found that, without influencing the expression of the p53 mRNA and protein, HIF-1α knockdown dramatically reduced the expression of the p21 mRNA and protein in CML cells, indicating that HIF-1α can control the proliferation of CML cells by controlling the expression of p21. Young-Suk Cho et al. [67] revealed that an intradermal injection of HIF-1α siRNA reduced p21 expression and induced skin hyperplasia in rat epidermis. All of the above evidence suggests that there is a strong link between HIF-1α and the p21 protein in apoptosis.

From our results, it was found that, compared with WT mice, the hippocampal tissues of the KO mice showed reduced VHL expression, elevated HIF-1α expression, and reduced p21 expression, which was further verified through RT-PCR experiments and found to be consistent with the transcriptomic results. We hypothesized that the reduction in VHL expression in the hippocampus of the scn1a KO mice may have impeded the proper ubiquitination and degradation of HIF-1, leading to an excessive accumulation of HIF-1α, which, in turn, targeted and modulated the expression of p21, ultimately resulting in apoptosis or cell cycle halt. To summarize, we found that the VHL/HIF-1α/p21 signaling pathway might be involved in hippocampal neuronal apoptosis in *Scn1a* KO mice.

## 4. Materials and Methods

### 4.1. Animals

The experimental animals were C57BL/6J female wild mice (provided by the Laboratory Animal Center of Ningxia Medical University) and 129S6/SvEvTac Scn1a (*Scn1a*
^+/−^) male mice (provided by Prof. Longjun Wu, now at the Mayo Clinic Graduate School of Biomedical Sciences and Department of Neurology, Rochester, MN, USA). The symptoms of Scn1a^+/−^ mice are strain-dependent [68]. It has been reported that mice carrying the 129S6/SvEvTac strain (129. Scn1a^+/−^) deletion lack a distinct phenotype and live normal lives. Nevertheless, when these mice are crossed with C57BL/6J mice, the resulting F1 generation displays a severe phenotype associated with Dravet syndrome [18,69]. The seizures in early Scn1a^+/−^ mice have been found to be influenced by both temperature and age, with almost all P20–22 and P30–46 mSMEI experiencing myoclonic seizures, which are followed by systemic seizures brought on by an elevated core body temperature. Spontaneous seizures have only been seen in mice aged P32 or older. It has also been indicated that an elevated temperature alone can induce seizures in patients [70].

All F1-generation mice (4–5 weeks) were utilized for the experiment. The mice were kept in a room with a 12 h light-and-dark cycle and humidity levels maintained at 50–60%. Every experiment was authorized by the Committee on Ethics for Experiments with Animals at the Experimental Animal Center at Ningxia Medical University. The animal study protocol was approved by the Medical Ethics Review Committee of Ningxia Medical University (approval no. 2022-G119; approval date 10 March 2022).

### 4.2. Mouse Genotype Identification

DNA was extracted from samples such as the F1-generation mice’s tails and nails (Tiangen Biotech, China). The concentration of the isolated DNA was determined before amplifying the DNA fragments using a PCR instrument. The primers were as follows: typical primer: 5′-AGTCTGTACCAGGCAGGAACTTG-3′; primer for wild-type reverse: 5′-CCCTGAGATGTGGGTGAATAG-3′; and primer for mutant reverse: 5′-AGACTGCCTTGGGAAAAGCG-3′ (Sangon Biotech, Shanghai, China). After the addition of the Taq enzyme, PCR experiments were conducted. The PCR reaction conditions included 30 PCR cycles—94 °C (30 s), 55 °C (30 s), and 72 °C (1 min)—followed by an initial denaturation at 94 °C (3 min). Finally, an extension at 72 °C (5 min) was performed. Agarose gel electrophoresis was performed using 6 μL of the PCR product, and the results were observed using a chemiluminescence gel imaging system.

### 4.3. Brain Electrodes

Isoflurane anesthetic was used on the mice, who were secured in a stereotactic frame to ensure that their heads were aligned flush from left to right. Following the application of iodophor to cleanse the scalp, the skull was incised at the midline, and the periosteum and associated connective tissues were gently removed to expose the anterior and posterior fontanelle and Bregma points. Electrodes were then implanted using a stereotactic device according to the method described in a previous study [19].

### 4.4. Hyperthermia-Induced Seizures in Mice

We subjected 10 F1-generation KO mice and 10 WT mice to heat in order to induce seizures. A temperature probe was fully inserted into the rectums of the mice and secured with tape, allowing them to move freely in a bucket. The length of the temperature probe was adjusted to ensure proper movement. The temperature was increased by 0.5 °C [71] every two minutes. If the mice exhibited seizures of grade 4 or higher, the heating lamp was immediately switched off, and the mice were subjected to moderate cooling in order to prevent sudden death from epilepsy due to excessive temperature. If the temperature reached 42.5 °C and the KO mice did not experience seizures, they were maintained at 42.5 °C for 5 min; if no seizures occurred, then the mice were excluded as experimental subjects. We induced seizures in the *Scn1a* KO mice five times, once a day, and samples were collected after the last seizure. A system for acquiring and processing signals in the experiments (BL-420 N, Techman) was utilized to observe and record the behavior and electroencephalograms of the epileptic seizures in the mice.

### 4.5. Nissl Staining

The mice were anesthetized using isoflurane. The thoracic cavities of the mice were exposed, and a solution of 0.9% saline was perfused from the left ventricle, followed by perfusion with 4% paraformaldehyde after washing out the blood. After the completion of perfusion, the mice’s entire brains were removed, and they were preserved for 12 to 24 h in 4% paraformaldehyde and then dehydrated with an alcohol gradient before being paraffin-embedded. The entire brains of the mice were cut into coronal sections, then baked in a 56-degree heating machine for 2 h to melt the wax. The sections were then sequentially immersed in xylene and gradient alcohol to remove the wax, followed by immersion in a toluidine blue solution for 30 min and glacial acetic acid for differentiation. Finally, the sections were dehydrated in a gradient manner to seal them. After the stained sections were dried, a Leica DM6 fluorescent microscope (Leica camera AG, Wetzlar, Germany) was used to capture images, and ImageJ was used for analyses.

### 4.6. TUNEL Staining

The paraffin embedding and dewaxing processes followed the same procedure described above. On the basis of the kit instructions (KeyGEN Biotech, Jiangsu, China), the samples were first treated with a ProteinaseK working solution, and then 50 µL of a TdT enzyme reaction solution was added to each sample. After that, the samples were incubated for sixty minutes at 37 °C in a warm box. Each sample was then exposed to 50 µL of a Streptavidin–TRITC labeling solution and incubated for thirty minutes at 37 °C in a humid box. Finally, the samples were sealed with DAPI sealing tablets after the dye had dried. A Leica DM6 fluorescent microscope (Leica camera AG, Wetzlar, Germany) was used to capture images, and ImageJ was used for analyses.

### 4.7. Omics Sample Preparation

Following the induction of heat-induced seizures in the mice, as described in Section 4.4, hippocampal tissues were carefully dissected from the ice using sterilized surgical instruments. After rinsing with 0.9% saline, the tissues were placed into pre-chilled cryotubes and immediately submerged in liquid nitrogen. Subsequently, the cryotubes were transferred to −80 °C ultra-low-temperature freezers for storage. Finally, the samples were packed in dry ice and sent for analysis.

### 4.8. Proteomic Experiments

#### 4.8.1. Protein Extraction and SDS–Polyacrylamide Gel Electrophoresis

The frozen hippocampal tissue was thoroughly pulverized in liquid nitrogen and, after that, a sufficient amount of the material was placed in a 1.5 mL centrifuge container. Then, 200 μL of pre-cooled SDS lysate (1% SDS, 50 mM Tris, PH8.1, sodium pyrophosphate, β-glycerophosphate, EDTA) (Beyotime, Shanghai, China,), a phosphatase inhibitor (Roche, Beijing, China), and a protease inhibitor (Amresco, Solon, OH, USA) were added to the tube. The solution was pulverized via ultrasonic waves on ice (power 80W, ultrasonic 1.0 s, off 1.0 s, total 2 min, repeated twice), and the mixture was centrifuged for 10 min (12,000 rpm, 4 °C). The supernatant was the total protein solution of the hippocampus. A BCA assay (Thermo Scientific, Emeryville, CA, USA) was used to measure each sample’s concentration.

For each sample, 10 μg of protein was extracted and separated using 12% SDS-PAGE. The separated gels were stained with Caumas Brilliant Blue (eStain LGProtein Stainer, GenScript Biotechnology Co., Ltd., Nanjing, China). A completely automated digital gel image analysis system was used to image the stained gel (Tanon, Shanghai, China).

#### 4.8.2. Trypsin Digestion and Peptide Labeling

Based on the determined protein concentration, 50 μg of protein was extracted from each sample, and this was then divided among the several groups of samples and adjusted to the same volume and concentration. Next, 5 mM dithiothreitol (DTT, Titan, Shanghai, China) was added to each sample (55 °C, 30 min); then, it was cooled to room temperature on ice, and the same volume of 10 mM iodoacetamide (IAA, Sangon, Shanghai, China) was added. The protein was precipitated by adding a 6-fold volume of acetone (OKA, Shanghai), and the sample was left for more than four hours at −20 °C or overnight. Then, the precipitate was collected via centrifugation at 4 °C, 8000× *g* for 10 min, and the acetone was evaporated for 2–3 min (acetone precipitation was used to remove the SDS). Next, 100 μL of TEAB (200 mM) (Sangon, Shanghai, China) was added to redissolve the precipitate, and 1/50 of the sample mass was added to 1 mg/mL of Trypsin–TPCK (Hualishi, Beijing), followed by digestion at 37 °C overnight. The samples were lyophilized and stored at −80 °C.

Next, 50 μL 100 mM TEAB (tetraethylammonium bromide, PH = 6.14, Sangon, Shanghai, China) was added to each lyophilized sample, followed by a TMT reagent (TMTsixplex™ 6-plex, Thermo Fisher, Waltham, MA, USA), which was cooled to room temperature. Then, 88 μL anhydrous acetonitrile was added, swirled for 5 min, and centrifuged. Then, 41 μL TMT reagent was added to the sample (room temperature, 1 h). The reaction was terminated by adding 8 μL 5% hydroxylamine (15 min), and the sample was freeze-dried and stored at −80 °C.

#### 4.8.3. Reversed-Phase (RP) Separation

Reversed-phase (RP) separation was performed using an Agilent Zorbax Extend RP column (5 μm, 150 mm × 2.1 mm) on a 1100 HPLC System (high-pH separation liquid chromatograph, Aglient, Beijing, China). RP gradient chromatography separation was achieved using mobile phases A (containing 2% acetonitrile (ACN, Thermo Fisher, Waltham, MA, USA)) and B (containing 90% acetonitrile). In this step, the mixed sample was chromatographically separated into 15 components for subsequent testing. The gradient elution conditions were 0~8 min, 98% A; 8~8.01 min, 98%~95% A; 8.01~48 min, 95~75% A; 48~60 min, 75~60% A; 60~60.01 min, 60~10% A; 60.01~70 min, 10% A; 70~70.01 min, 10~98% A; 70.01 to 75 min, 98% A. The tryptic peptides were separated at a flow rate of 300 µL/min and monitored at 210 nm. The samples were collected after 8–60 min, the eluent was collected into centrifuge tubes 1–15 at 1-minute intervals in sequence, and then the samples were collected repeatedly according to the sequence 1 → 15. The samples were collected, vacuum freeze-dried, and evacuated before being frozen and kept ready for mass spectrometry analyses.

#### 4.8.4. Mass Spectrometry Analyses

All analyses were performed using a Q-Exactive mass spectrometer (Thermo Fisher, Waltham, MA, USA) equipped with a Nanospray Flex source (Thermo, Waltham, MA, USA). The samples were loaded at a flow rate of 2 µL/min onto a pre-column, Acclaim PepMap100 100 μm × 2 cm (RP-C18, Thermo Fisher, Waltham, MA, USA) (loading time 3 min), and then they were separated at a flow rate of 300nL/min onto an analytical column, Acclaim PepMap RSLC, 75 μm × 50 cm (RP-C18, Thermo Fisher, Waltham, MA, USA). The linear gradient was 75 min (0~40 min, 5–28% B; 40~60 min, 28–42% B; 60~65 min, 42–90% B; 60~65 min, 90% B; mobile phase A = 0.1% formic acid (FA, Thermo Fisher, Waltham, MA, USA) in water and B = 0.1% FA in ACN). The first MS resolution was set to 60,000, the automatic gain control value was set to 1 × 10^6^, and the maximum injection time was 50 ms. The mass spectral scan was set to a full-scan charge-to-mass ratio *m*/*z* range of 350–1500; all MS/MS plot acquisitions were accomplished using high-energy collisional cleavage in the data-dependent positive-ion mode, with the collisional energy set to 36. The MS/MS resolution was set to 30,000, the automatic gain control was set to 1 × 10^5^, and the maximum accumulation time of ions was set to 80 ms; the dynamic exclusion time was set to 30 s.

#### 4.8.5. Bioinformatics Analyses

Proteome Discover 2.4 software (Thermo Fisher, Waltham, MA, USA) was used to process the raw data (the specific parameter settings are shown in Table 2), and the data were compared with the uniprot reviewed_yes+taxonomy_10090.fasta database. After the original data were retrieved from the database, trusted proteins were screened according to Score Sequest HT > 0 and unique peptide ≥ 1, with the blank values removed. The statistical significance of the data was determined using Student’s *t*-test. Proteins with a *p*-value < 0.05 and a fold change >1.2 (upregulated) or a fold change <1/1.2 (downregulated) were defined as differentially expressed. Percolator was used to determine the False Discovery Rate (FDR) of a spectrum match. To perform GO/KEGG functional enrichment analyses, we used species proteins as the background list and identified differential proteins as the candidate list. We then used a hypergeometric distribution test to determine the significance of function enrichment in the differential protein list, followed by correction using Benjamini and Hochberg’s multiple tests to obtain the FDR. The KEGG database was used for pathway analyses of the DEPs. The analyses and spectrum matching of the proteomes were carried out by OE Biotech Co. Ltd., located in Shanghai, China (methodological details can be found in Appendix A).

### 4.9. Transcriptomic Experiments

As directed by the manufacturer, the total RNA was extracted using the TRIzol reagent (Invitrogen, CA, USA). RNA quantity and purity were assessed with a NanoDrop 2000 spectrophotometer (Thermo Scientific, Waltham, MA, USA). An Agilent 2100 Bioanalyzer was used to evaluate the integrity of the RNA (Agilent Technologies, Santa Clara, CA, USA). A VAHTS Universal V6 RNA-seq Library Prep Kit was then used to build libraries in accordance with the manufacturer’s instructions. An RNA-seq analysis was conducted by OE Biotech Co., Ltd. (Shanghai, China).

The libraries were sequenced using an Illumina Novaseq 6000, and 150 bp paired-end reads were generated. Fastp was initially used to process the raw data in the fastq format, and low-quality reads were eliminated to produce clean reads. The clean reads were mapped to the reference genome using HISAT2 (Appendix A).

Annotation files and reference gene sequences were used as a database (genome database: https://ftp.ncbi.nlm.nih.gov/genomes/all/GCF/000/001/635/GCF_000001635.27_GRCm39/GCF_000001635.27_GRCm39_genomic.fna.gz (accessed on 15 March 2023); mRNA database: https://ftp.ncbi.nlm.nih.gov/genomes/all/GCF/000/001/635/GCF_000001635.27_GRCm39/GCF_000001635.27_GRCm39_rna.fna.gz (accessed on 15 March 2023). By comparing the sequence similarity, the expression level of each protein-coding gene in each sample was determined. The FPKM of each gene was calculated, and the read counts of each gene were obtained using HTSeq-count. R (v3.2.0) was used to conduct a PCA analysis in order to assess the biological duplication of the samples. A differential expression analysis was performed using DESeq2. The difference multiples were calculated, and NB (the method of the negative binomial distribution test) was used to test the significance of the differences. A q-value <0.05 and a fold change >2 or a fold change <0.5 were set as the thresholds for significantly differentially expressed genes (DEGs). A hierarchical cluster analysis of the DEGs was performed using R (v 3.2.0) to demonstrate the expression patterns of the genes in the different groups and samples. A radar map of the top 30 genes was drawn to show the expression of the upregulated and downregulated DEGs using R packet ggradar.

Based on the hypergeometric distribution, GO [72] and KEGG [73] pathway enrichment analyses of the DEGs were conducted to identify the significantly enriched terms using R (v 3.2.0), and column diagrams, chord diagrams, and bubble diagrams of the significantly enriched terms were drawn using R (v 3.2.0).

### 4.10. Western Blot Tests

The hippocampal tissues of the F1-generation mice were excised immediately following the heat-induced seizures. A whole-protein extraction kit was utilized to extract the proteins, and the protein content was assessed using a BCA kit (KeyGEN Biotech, Jiangsu,China). The protein samples underwent SDS-PAGE gel electrophoresis in equal quantities (70 V, 30 min–110 V, 60 min), followed by membrane transfer at 300 mA for 60 min and blocking in 5% BSA for 1h. Next, the antibodies were incubated at 4 °C for the entire night on a shaker. The next day, the membrane was treated with sheep and rabbit antibodies for one hour at room temperature on a shaking bed. After the Western blot tests, chemiluminescence images were created, taken, and stored, and Image J (Fiji × 64) software was used to evaluate the data.

The primary antibodies were as follows: anti-Vhl mouse monoclonal antibody (sc-135657, 1:1000, Santa Cruz Biotechnology, Shanghai, China), anti-Aqp1 rabbit polyclonal antibody (AF5231, 1:1000, Affinity Biosciences, Jiangsu, China), anti-Ace rabbit polyclonal antibody (24743-1-AP, 1:1000, Proteintech, Wuhan, China), anti-Folr1 rabbit polyclonal antibody (23355-1-AP, 1:2000, Proteintech, Wuhan, China), anti-CDKN1A (p21) mouse monoclonal antibody (sc-6246, 1:1000, Santa Cruz Biotechnology, Shanghai, China), anti-HIF-1α polyclonal antibody (AF1009, 1:1000, Affinity Biosciences, Jiangsu, China), Dylight 800 goat anti-rabbit IgG (A23920, 1:500, Abbkine Scientific Co., Ltd., Wuhan, China), and Dylight 800 goat anti-mouse IgG (A23910, 1:500, Abbkine Scientific Co., Ltd., Wuhan, China).

### 4.11. Real-Time Quantitative Polymerase Chain Reaction Experiments

In compliance with the RNA extraction kit’s instructions (Tiangen Biotech, Beijing, China), the total RNA was extracted from the mouse hippocampal tissue. The TaKaRa RR037A reagent was then used to reverse-transcribe the RNA into cDNA, and RT-qPCR was conducted using FastFire qPCR premix. The primer sequences are depicted in Table 3, and they were all obtained from Shanghai Bioengineering Company. iQ5 software (v 2.1,Bio Rad Laboratories, Inc. Hercules, CA, USA) was employed for PCR amplification. The amplification conditions were as follows: 39 cycles in total—first denaturation at 95 °C for 15 min, then denaturation at 95 °C (10 s), annealing at 61.5 °C (20 s), and extension at 72 °C (30 s). The 2^−ΔΔCt^ technique was utilized to determine the relative expression of the genes. Three iterations of these experiments were conducted. The primer sequences are shown in Table 3.

### 4.12. Statistical Analyses

Prism9.5.1 was used to perform statistical analyses on all of the data. The two groups were compared using Student’s *t*-test, and the measurement data are shown as the mean ± standard deviation. Each experiment was performed three times, and the results were considered statistically significant at *p* < 0.05 (* *p* < 0.05, ** *p* < 0.01, *** *p* < 0.001, **** *p* < 0.0001).

Genes, protein interaction networks, and inter-pathway relationship networks were obtained from the KGML file, and they were visualized and interpreted using the networkx package (V2.5) with Python (V3.8.8).

## 5. Conclusions

Initially, we detected neuronal cell death in the hippocampus of *Scn1a* KO mice through Nissl and TUNEL staining. To delve deeper into the underlying mechanisms of neuronal cell death, we integrated proteomic and transcriptomic techniques and identified the potential involvement of the VHL/HIF-1α/P21 signaling pathway.

## Figures and Tables

**Figure 1 ijms-25-04457-f001:**
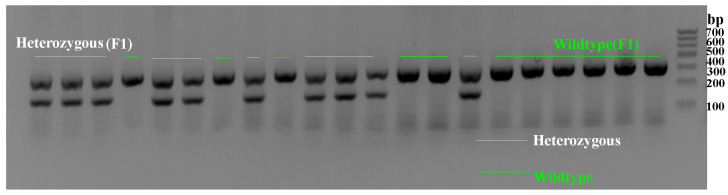
*Scn1a* KO (heterozygous) mice and WT mice were identified using PCR.

**Figure 2 ijms-25-04457-f002:**
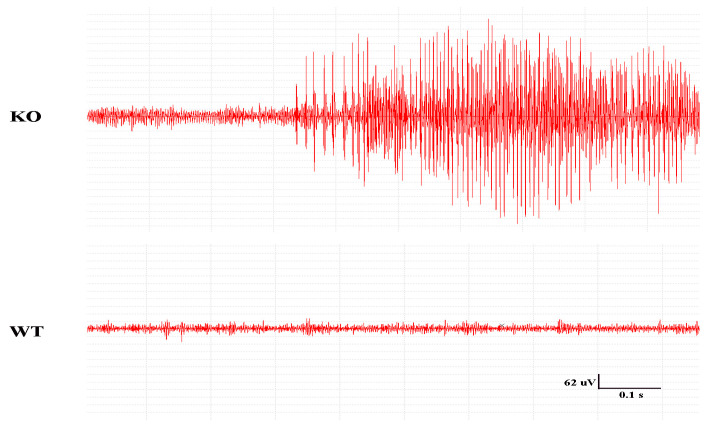
A representative EEG recording of seizure activity in the KO and WT mice.

**Figure 3 ijms-25-04457-f003:**
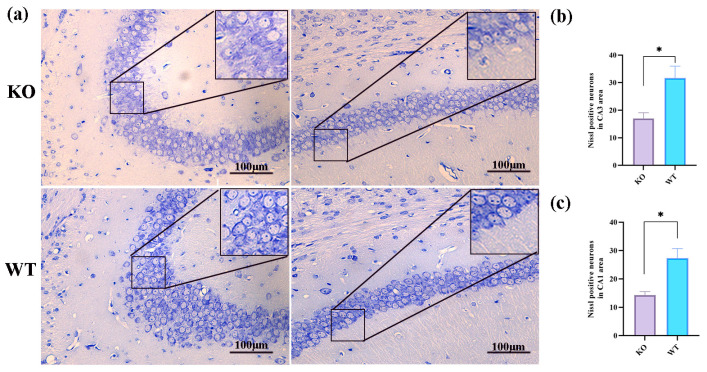
Compared with WT mice, KO mice exhibited a loss of neurons in the hippocampus. (**a**) Representative Nissl staining images of CA3 and CA1 in mouse hippocampal regions. (**b**) Statistical plot of the number of Nissl staining-positive cells in the CA3 region. (**c**) Statistical plot of the number of Nissl staining-positive cells in the CA1 region. (**d**) Representative TUNEL staining images of CA3 and CA1 in the hippocampi of the mice. (**e**) Statistical plot of the number of TUNEL staining-positive cells in the CA3 region. (**f**) Statistical plot of the number of TUNEL staining-positive cells in the CA1 region. All data are expressed as the mean ± SEM (n = 3). * *p* < 0.05 and ** *p* < 0.01. Scale bar = 100 μm.

**Figure 4 ijms-25-04457-f004:**
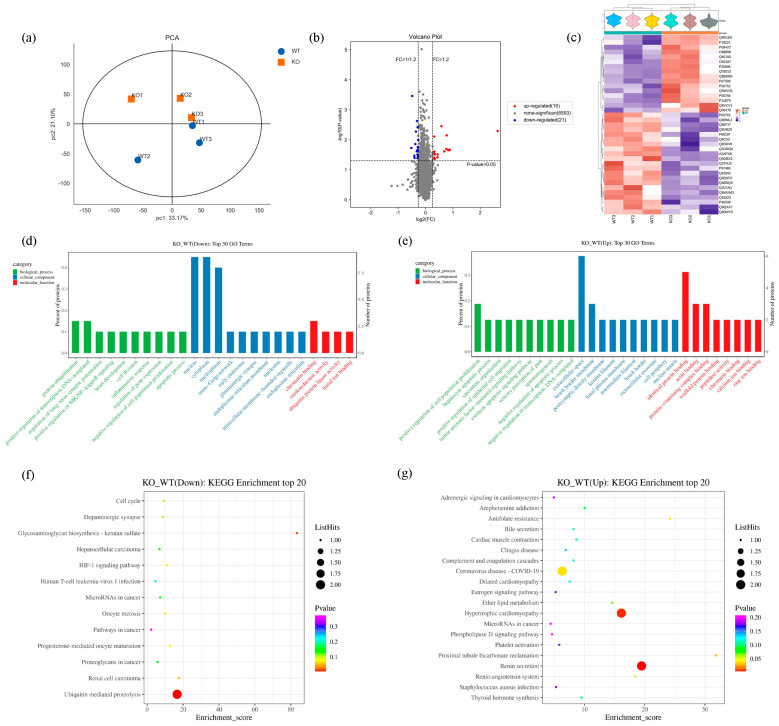
Details of the protein identification and multivariate statistical analyses grounded in proteomic research. (**a**) Principal component analysis (PCA). (**b**) Volcano map of differentially expressed proteins. (**c**) Clustering heatmap of differentially expressed proteins. (**d**) Gene Ontology (GO) annotation classification statistics chart of the top 30 downregulated differential proteins. (**e**) Gene Ontology (GO) annotation classification statistics chart of the top 30 upregulated differential proteins. (**f**) KEGG enrichment analysis bubble map of the top 20 downregulated differential proteins. (**g**) KEGG enrichment analysis bubble map of the top 20 upregulated differential proteins.

**Figure 5 ijms-25-04457-f005:**
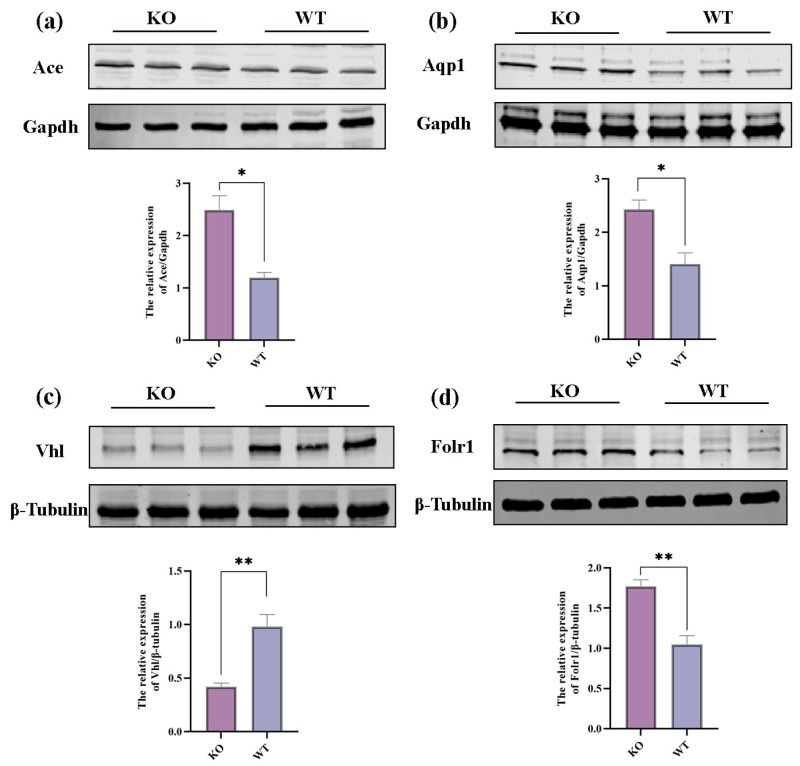
Validation of DEPs in proteomic data through WB experiments. (**a**–**d**) Representative Western blot images and statistical plots for Ace, Aqp1, Vhl, and Folr1. All data are expressed as the mean ± SEM (n = 3). * *p* < 0.05 and ** *p* < 0.01.

**Figure 6 ijms-25-04457-f006:**
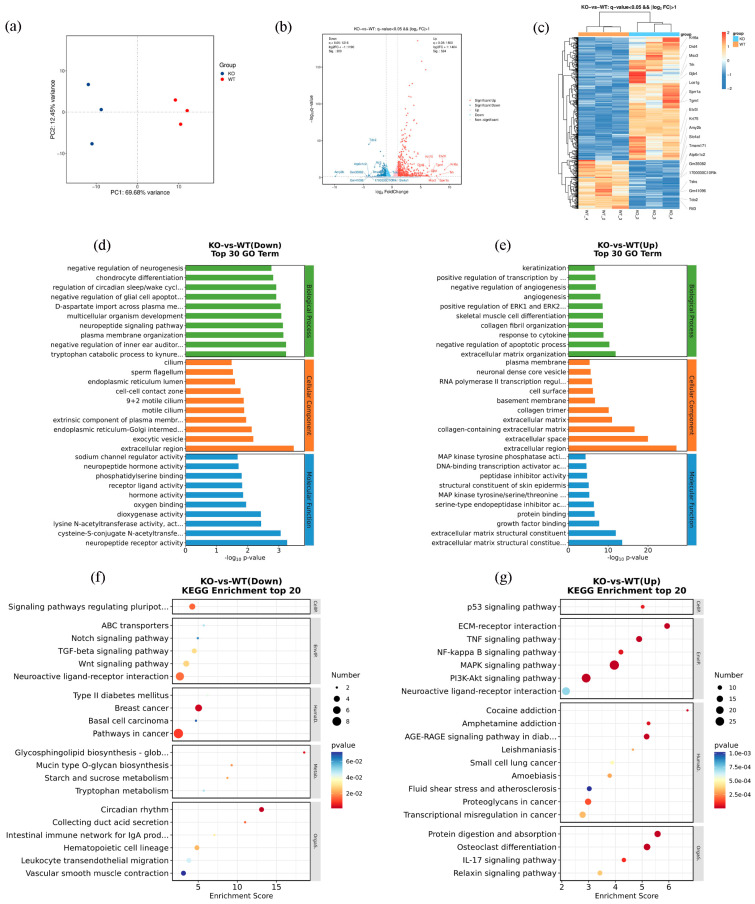
Details of the protein identification and multivariate statistical analyses grounded in transcriptome research. (**a**) PCA score plots of hippocampal tissue samples between KO and WT mice. (**b**) Volcano map of differentially expressed genes. (**c**) Cluster heatmap of differentially expressed genes. (**d**) GO enrichment map of downregulated differential genes. (**e**) GO enrichment map of upregulated differential genes. (**f**) KEGG enrichment map of downregulated differential genes. (**g**) KEGG enrichment map of upregulated differential genes.

**Figure 7 ijms-25-04457-f007:**
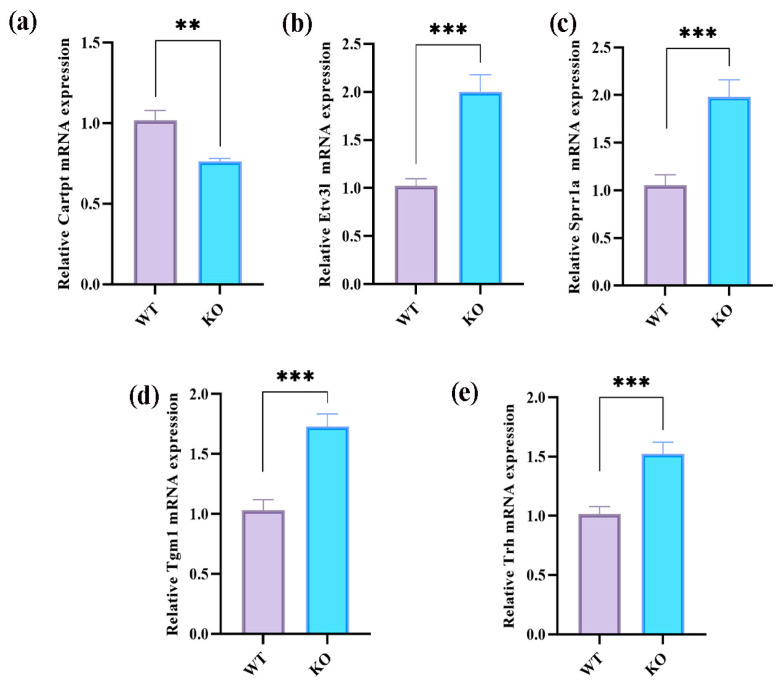
Validation of transcriptomic data through real-time quantitative polymerase chain reaction experiments. (**a**–**e**) Representative RT-PCR statistical plot for Cartpt, Sprrla, Tgm1, Trh, and Etv3l. All data are expressed as the mean ± SEM. ** *p* < 0.01 and *** *p* < 0.001.

**Figure 8 ijms-25-04457-f008:**
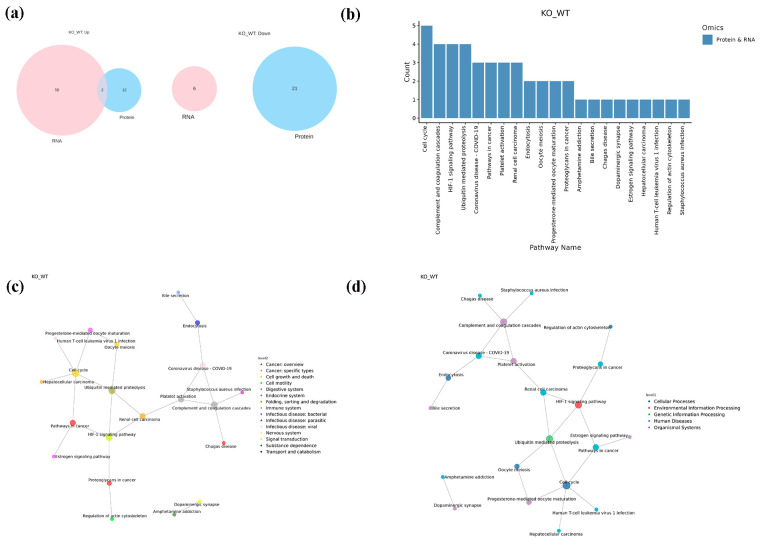
Integrated information from proteomic and transcriptomics analyses. (**a**) Venn diagram of proteomic and transcriptomic association analysis. (**b**) Distribution map of the top 30 associated elements. (**c**) KGML interaction network diagram: level 1. (**d**) KGML interaction network diagram: level 2.

**Figure 9 ijms-25-04457-f009:**
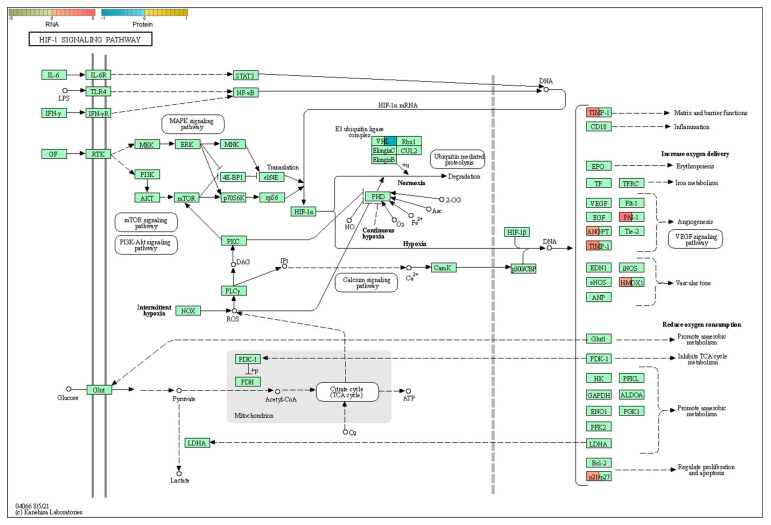
KEGG map (mmu04066). The boxes represent genes/proteins (genes on the left, and proteins on the right), and the circles represent metabolites. Color gradients are used to show specific expressions.

**Figure 10 ijms-25-04457-f010:**
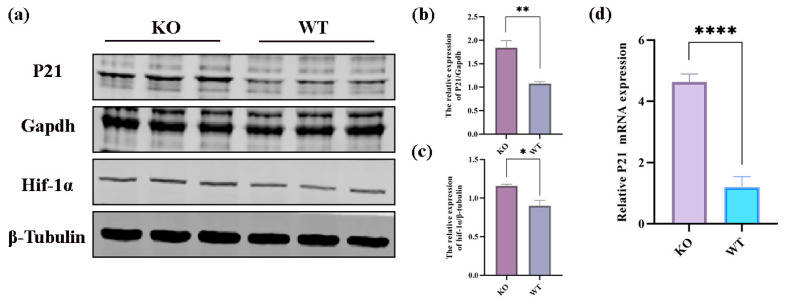
Expression of DGPs and DGEs in the HIF-1 signaling pathway. (**a**–**c**) Representative Western blot images and statistical plots of P21 and HIF-1α. (**d**) Representative RT-qPCR statistical plot of P21. All data are expressed as the mean ± SEM. * *p* < 0.05 and ** *p* < 0.01. **** *p* < 0.0001.

**Table 1 ijms-25-04457-t001:** Co-expression pathways and the proteins and genes involved.

Number	KEGG	Pathway Names	Genes	Proteins
1	mmu00533	Glycosaminoglycan biosynthesis– keratan sulfate	B4galt1	Chst2
2	mmu00565	Ether lipid metabolism	Plb1	Enpp2
3	mmu04066	HIF-1 signaling pathway	Angpt2;Cdkn1a;Hmox1;Serpine1;Timp1	Vhl
4	mmu04080	Neuroactive ligand–receptor interaction	Adm;Bdkrb2;C5ar1;Calcr;Drd1;Drd4;Edn3;F2rl2;Gal;Ghsr;Glra1;Hcrtr1;Npbwr1;Npsr1;Npy;Npy2r;Ntsr1;Penk;Prlr;Sstr2;Tacr1;Trh;Vgf	F2
5	mmu04110	Cell cycle	Cdkn1a;Gadd45a;Gadd45b;Gadd45g	Anapc7
6	mmu04114	Oocyte meiosis	Mapk12	Anapc7
7	mmu04120	Ubiquitin-mediated proteolysis	Socs3;Ubc	Anapc7,Vhl
8	mmu04144	Endocytosis	Hspa1a;Hspa1b	Folr1
9	mmu04261	Adrenergic signaling in cardiomyocytes	Creb3l1;Mapk12	Tnnt2
10	mmu04610	Complement and coagulation cascades	A2m;Bdkrb2;C4b;C5ar1;F2rl2;Plaur;Serpinb2;Serpine1;Serpinf2	F2
11	mmu04611	Platelet activation	Itga2;Mapk12;Mylk3	F2
12	mmu04728	Dopaminergic synapse	Creb3l1;Drd1;Drd4;Fos;Mapk12;Slc18a2;Th	Scn1a
13	mmu04810	Regulation of actin cytoskeleton	Bdkrb2;Fgf21;Itga2;Itga5;Myh9;Mylk3	F2
14	mmu04914	Progesterone-mediated oocyte maturation	Mapk12	Anapc7
15	mmu04915	Estrogen signaling pathway	Creb3l1;Fos;Hspa1a;Hspa1b;Jun;Krt14;Krt15;Rara	Krt18
16	mmu04918	Thyroid hormone synthesis	Creb3l1;Hspa5	Ttr
17	mmu04924	Renin secretion	Aqp1;Edn3;Kcnj2	Ace,Aqp1
18	mmu04964	Proximal tubule bicarbonate reclamation	Aqp1	Aqp1
19	mmu04976	Bile secretion	Aqp1;Hmgcr	Aqp1
20	mmu05031	Amphetamine addiction	Arc;Creb3l1;Drd1;Fos;Fosb;Jun;Slc18a2;Th	Arc
21	mmu05142	Chagas disease	Bdkrb2;Ccl2;Ccl3;Fadd;Fos;Jun;Mapk12;Serpine1;Tlr2	Ace
22	mmu05150	Staphylococcus aureus infection	C4b;C5ar1;Dsg1c;Fcgr2b;Fcgr4;Icam1;Krt14;Krt15	Krt18
23	mmu05166	Human T-cell leukemia virus 1 infection	Cdkn1a;Creb3l1;Egr1;Egr2;Fos;Fosl1;Icam1;Il1r2;Jun;Msx3;Zfp36	Anapc7
24	mmu05171	Coronavirus disease—COVID-19	C4b;C5ar1;Ccl2;Fos;Jun;Mapk12;Tlr2	Ace,F2
25	mmu05200	Pathways in cancer	Bbc3;Bdkrb2;Cdkn1a;Col4a1;Epor;Fadd;Fgf21;Fos;Frat2;Gadd45a; Gadd45b;Gadd45g;Hes5;Hmox1;Itga2;Jun; Nkx3-1;Pim1;Pmaip1;Ptgs2;Rara;Ret;Runx1;Wnt2;Wnt8b;Wnt9b	Vhl,F2
26	mmu05205	Proteoglycans in cancer	Cd44;Cdkn1a;Flnc;Itga2;Itga5;Mapk12;Plaur;Sdc1;Thbs1;Tlr2;Wnt2; Wnt8b;Wnt9b	Pdcd4
27	mmu05206	MicroRNAs in cancer	Cd44;Cdkn1a;Hmox1;Itga5;Mcl1;Pim1;Ptgs2;Spry2;Thbs1;Tnc	Pdcd4,Tp63
28	mmu05211	Renal cell carcinoma	Cdkn1a;Jun	Vhl
29	mmu05225	Hepatocellular carcinoma	Cdkn1a;Frat2;Gadd45a;Gadd45b;Gadd45g;Hmox1;Wnt2;Wnt8b; Wnt9b	Pbrm1
30	mmu05410	Hypertrophic cardiomyopathy	Itga2;Itga5	Ace,Tnnt2
31	mmu05414	Dilated cardiomyopathy	Itga2;Itga5	Tnnt2
32	mmu05415	Diabetic cardiomyopathy	Mapk12	Ace

**Table 2 ijms-25-04457-t002:** Mass spectrometry retrieval parameters.

Items	Settings
Static modification	TMT (N-term, K); carbamidomethyl (C)
Dynamic modification	Oxidation (M), acetyl (N-term)
Digestion	Trypsin
Instrument	Orbitrap fusion
MS1 tolerance	10 ppm
MS2 tolerance	0.02 Da
Missed cleavages	2
Database	uniprot-reviewed_yes+taxonomy_10090.fasta

**Table 3 ijms-25-04457-t003:** Sequences of the forward and reverse primers.

Gene	Forward	Reverse
Trh	5′-ACCTTGGCTGATGATGGCTCTG-3′	5′-CTTCCTCCTGGGCTGCTTCC-3′
Tgm1	5′-GTGGAACGACTGCTGGATGAAG-3′	5′-GACTAAGCCATTCTTGACGGACTC-3′
Sprr1a	5′-GCCAGCCTAAGGTGCCAGAG-3′	5′-GTATGGTGATGGAGTGACAGTTGAG-3′
Etv3l	5′-CAAGACCAAAGGCAAGAAGTTTACC-3′	5′-AGGCACCAAGGGCTGACAC-3′
Cartpt	5′-GTGCCCGTGCCCAGGAG-3′	5′-TCTTGAGCTTCTTCAGGACTTCTTG-3′
P21	5′-CCGTGGACAGTGAGCAGTTG-3′	5′-CCTCCAGCGGCGTCTCC-3′

## Data Availability

Data are contained within the article.

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
