# Peer review of "Integrating Proteomics and Transcriptomics Reveals the Potential Pathways of Hippocampal Neuron Apoptosis in Dravet Syndrome Model Mice"

_ijms, 2024, doi:10.3390/ijms25084457_

Round 1

Reviewer 1 Report

Comments and Suggestions for Authors

This study compared protein and transcriptomic profiles of WT and SCNIA KO mice following heat-induced seizures.

The manuscript is well-written and easy to follow. However I have concerns as listed below:

The choice of substrain can significantly alter disease phenotype. It is not clear what the phenotype of the F1 animals in this study looked like. Under methods there is a reference to what it looks like in another study but the phenotype of the animals in the current study, prior to hyperthermia-induced seizures, is not adequately described but would be relevant as the level of spontaneous seizures could impact on the outcome.

Not all animals experience seizure following hyperthermia, there are no details on for how long the animals were observed after the temp reaches a certain point – e.g in some studies if no seizures observed within 5 minutes the animal is considered seizure free.

Fig2 – is this a readout from one animal from each group? Can this be shown for all animals?

It is assumed that molecular changes following hyperthermia-induced seizures are a consequence of the seizure, but could they also be due to heat stress?

The background discusses neuronal apoptosis within the context of ongoing seizure activity (chronic), whereas the study looks at omic changes immediately following an induced seizure (acute) effects. This needs to be addressed in the discussion.

If the p-value for the proteomic results are adjusted for multiple testing it would seem that non are differentially expressed across groups. This might not be surprising as proteins are less variable than RNA transcripts. The PCA plot should be included for the proteomics. The fold change is very small, yet the heatmap would indicate that it is greater – the volcano plots on the heatmap should be labelled with the sample ID.

The spreadsheet S1 doesn’t show any downregulated proteins.

Fig5 – re 4 confirmed dysregulated proteins (Ace Aqp1 Vhl and Folr1) – it says these ‘were selected’ but no justification is given as to why they were selected – not the top 4 based on unadjusted p value or fold change.

There are bioinformatics methods for the integration of omics data e.g MOFA, MixOmics that could have potentially been applied. It seems here that the proteins and genes were included in the same enrichment analysis but it was not very clear to me exactly how this was done – was the protiens mapped to genes and all gene symbols uploaded to a tool?

The transcriptomics data indicated UP ‘negative regulation of apoptotic process’ and Down had negative regulation of glial cell apoptosis – more discussion or focus on this might be warranted given the objective of the study.

The authors selected to focus on the HIF-1 signalling pathway, which seems to be based on the expression of just 4 genes - Angpt2;Cdkn1a;Hmox1;Serpine1;Timp1 and the protein Vhl. Other genes were more differentially expressed (e.g. Krt6a) is there any relationship between the most dysregulated genes and the four genes in the HIF-1 pathway?

it should be clarified that P21 is another name for CDKN1A – if that is the case.

The abstract mentions the drug liraglutide which I think is an anti-diabetic drug – how does this relate to the finding in this study i.e does this drug impact on the HIF-1 pathway?

Minor editing

It should be clarified what WB experiment is

A space between sentences is often missing

Author Response

Thank you for your help in reviewing the manuscript and making useful suggestions. We have revised the manuscript according to your suggestions.

Reviewer 2 Report

Comments and Suggestions for Authors

Presented study aims to elucidate the molecular basis of Hippocampus neuron apoptosis in of Scn1a KO mice, with the dravet syndrome, the animal model of epilepsy. Authors confirmed that F1 heterozygote mice for sure demonstrate an epileptic behaviour and can be used for further molecular studies, however, the details of proteomic, transciptomic and bioinformatics analysis are just partially available. My concerns are listed below:

1. Dravet syndrome (in the abstract) is not defined, same as KGML.

2. Differential abundance - how it was done?

3. GO should be capital, not 'go' (lines 119-136)

4. Why enrichment of COVID is not discussed?

5. How four proteins for WB was selected?

6. No numbers for the transcriptomic analyses, how many genes in total were identified? For proteins the ~7000 is stated.

7. Why the description of volcano plot appears in transcriptomics, not in proteomics analysis?

8. Chord diagram is not readable even when enlarged, the other figures like volcano plots, enrichment, etc, are very small and cannot be read in printed format. Figure 4, H shows SWISS PRot IDs for genes - gene name would be much more informative.

9. Why genes were selected for RT-PCR randomly?

10. Correlation analysis of DE data in proteomics and transcriptomics should be shown even the if the correlation is low

11. Is there an overlap between DEGs and DEPs? how big is it?

12. No device is mentioned in the proteomics method, actually the only statement which is needed "The proteome data was collected and analyzed by OE Biotech Co., Ltd., located in 454 Shanghai, China."

13. No details for bioinformatics analysis is presented, e.g for KGML? If it was done with R - which packages?

Comments on the Quality of English Language

I'd recommend additional proofreading for English, some sentences are confusing.

Author Response

Thank you for your help in reviewing the manuscript and making useful suggestions. We have revised the manuscript according to your suggestions.

Journal:IJMS (ISSN 1422-0067)

Manuscript ID:ijms-2900898

Title:Integrated Proteomics And Transcriptomics Reveal The Potential Pathway Of Hippocampal Neuron Apoptosis In Dravet Syndrome Model Mice

Reviewer#2 Presented study aims to elucidate the molecular basis of Hippocampus neuron apoptosis in of Scn1a KO mice, with the dravet syndrome, the animal model of epilepsy.  Authors confirmed that F1 heterozygote mice for sure demonstrate an epileptic behaviour and can be used for further molecular studies, however, the details of proteomic, transciptomic and bioinformatics analysis are just partially available. My concerns are listed below:

Response: Thank you very much for the positive comments.

Comments 1:Dravet syndrome (in the abstract) is not defined, same as KGML.

Response:Thank you very much for your professional advice.We apologize for the ignored of description.We have been added the defined Dravet syndrome and KGML in revised manuscript.

Comments 2: Differential abundance - how it was done?

Response:Thank you very much for your professional advice.We apologize for the ignored of description. Proteomics Differential protein screening based on FC and p values; Screening conditions up-regulated differential protein FC≥1.2 and p<0.05, down-regulated differential protein FC≤1/1.2 and p<0.05. Difference analysis of transcriptome was performed using DEseq2 and screened according to T-test difference test, FC≥2 or ≤0.5, P (or q < 0.05).

Comments 3: GO should be capital, not 'go' (lines 119-136)

Response:Thank you very much for your professional advice.We apologize for the error description. We have corrected in revised manuscript.

Comments 4: Why enrichment of COVID is not discussed?

Response:Thank you very much for your professional advice.We apologize for the ignored of description.We have added in revised manuscript.

Comments 5:How four proteins for WB was selected?

Response:Thank you very much for your professional advice.We apologize for the ignored of description.The four proteins were selected randomly for the purpose of verifying the results of proteomic analysis through WB experiment. Although the results were not exactly the same, the expression trend of DEPs was the same, which confirmed the consistency of the two methods.It is also to avoid errors caused by individual willingness to choose proteins with large differences for verification

Comments 6:No numbers for the transcriptomic analyses, how many genes in total were identified?  For proteins the ~7000 is stated.

Response:Thank you very much for your professional advice. We apologize for the ignored of description.We have added it in the revised manuscript Supplementary material.(Supplementary S4).

Comments 7:Why the description of volcano plot appears in transcriptomics, not in proteomics analysis?

Response:Thank you very much for your professional advice. We apologize for the ignored of description.We have added a description of the volcano map of proteomics in the manuscript.

Comments 8:Chord diagram is not readable even when enlarged, the other figures like volcano plots, enrichment, etc, are very small and cannot be read in printed format.  Figure 4, H shows SWISS PRot IDs for genes - gene name would be much more informative.

Response:Thank you very much for your professional advice. We apologize for the error of description.We reuploaded the picture in the manuscript.The 37 different ID names on the right of the figure represent 37 different proteins.The corresponding protein name can be found in the Supplementary S2.

Comments 9: Why genes were selected for RT-PCR randomly?

Response:Thank you very much for your professional advice. We apologize for the ignored of description. Genes were randomly selected and the transcriptomic analysis results were verified by RT-qPCR. Although the results were not identical, the expression trend of DEGs was the same, confirming the consistency of the two methods. This is also to avoid the error caused by the willingness of individuals to select widely different proteins for verification

Comments 10:Correlation analysis of DE data in proteomics and transcriptomics should be shown even the if the correlation is low

Response:Thank you very much for your professional advice. We apologize for the ignored of description.We analyzed the correlation between proteome and transcriptome in the manuscript.

Comments 11:Is there an overlap between DEGs and DEPs? how big is it?

Response:Thank you very much for your professional advice. We apologize for the ignored of description.There are only 2 of the up-regulated omics overlap, with no overlap seen in the down-regulated omics.The connection between the proteome and transcriptome was found to be limited,aligning with findings from past research on discrepancies between the two.

Comments 11:No device is mentioned in the proteomics method, actually the only statement which is needed "The proteome data was collected and analyzed by OE Biotech Co., Ltd., located in 454 Shanghai, China."

Response:Thank you very much for your professional advice. We apologize for the ignored of description.We added some experimental equipment for proteomics to the manuscript.

Comments 12:No details for bioinformatics analysis is presented, e.g for KGML?  If it was done with R - which packages?

Response:Thank you very much for your professional advice. We apologize for the ignored of description.We have added in the revised manuscript.

Reviewer 3 Report

Comments and Suggestions for Authors

Kong et al. submitted an interesting paper concerning the analysis of the potential molecular background of epilepsy.

  •  I would like to know more about the obtained results, but the size and format of some important Figures are completely unreadable in pdf printed format: Figures 4, 6, 8, 9, 19B, C. If it is difficult to present all data, maybe the authors should choose the most important, enlarge, and the rest put into the supplementary data? 
  • Figure 1 must be corrected and presented blot converted to Figure - delete the unnecessary background, put the titles above the photo, etc. 
  • Wildtype- Wild Type 
  • pcr - PCR
  • There are several punctuation errors, a lack of spaces (xxmin - xx min, xxul - xx ul etc.
  • the ref citation within the text does not follow the IJMS template, for example [11; 12; 13;] - [11-13]
  • In the Western blot method please provide the cat. no. of all Abs along with used dilutions
  • Fig. 5A - the graph does not correspond with the presented blot, there are no visible such significant changes in Ace in KO and WT?
  • unify the style of used abbreviations, for example, there are: HIF1a, HIF-1a, Hif-1a, HIF-1a
  • Table 2, please indicate the 5' and 3' in the sequences of the primers
  • The references style does not follow IJMS template
  •  

Comments on the Quality of English Language

 Extensive editing of English language required

Author Response

(The authors gave the same response as above.)

Reviewer 4 Report

Comments and Suggestions for Authors

The study aimed to investigate the potential molecular mechanisms of neuronal apoptosis in a mouse model of Dravet syndrome using integrated proteomics and transcriptomics approaches. The results showed increased neuronal cell death and differential expression of proteins and genes related to apoptosis, synaptic functions, neurotransmission and cell cycle in the hippocampus of Scn1a KO mice. Overall, the study addressed an important topic but some aspects of the methodology and results need clarification:

1. In the introduction, the authors should further explain and justify conducting both proteomic and transcriptomic analyses given the overlapping purpose of the two approaches.

2. Figure 1 could be improved by clearly labeling all lanes for easier interpretation. 

3. Figure 3d needs to specify what the color represents for clarity.

4. Section 2.4:

a. A list of the 7,155 identified proteins should be provided as stated.

b. Column L-Q in Supplementary Table S1 should be defined as the normalized intensity measures for easier understanding.

c. Line 108 refers to Supplementary Table S1 not S2.

d. Results of PCA for the proteomic data are missing.

e. Figures 4e and f could be improved by sorting GO terms by enrichment score rather than alphabetically for clearer interpretation.

5. More methodological details are needed for the transcriptomic analysis, such as number of reads, mapping rates, number of genes after assembly etc. 

6. In line 153-158 and Figure 6, clarify that the colored dots indicate the top 10 most up/downregulated genes as there appear to be more than 20 dots. Also, Figure 6b is redundant with 6c.

7. In line 172-173 and Figure 6e/f, "bubbles" should be clarified or corrected as the figures do not contain bubbles.

8. Section 4.8 is missing several important methodological details for the proteomic sample preparation and data analysis:

a. Line 430 refers to "SDS lysate" but does not specify the composition of the SDS lysis buffer used (e.g. SDS concentration, additives) or the volume added to the sample.

b. Lines 438 and 443 refer to two different LC columns - an Agilent Zorbax Extend RP column and an Acclaim PepMap RSLC, 75μm×50cm (RP-C18, Thermo Fisher) column. Please clarify which column was used and for which step (trap vs analytical).

c. Lines 444-445 describe loading the sample onto a "pre-column Acclaim PepMap100, 100μm×2cm" followed by the "analytical column". Please confirm the order of operations and clarify if the pre-column was used for desalting/concentration prior to analysis.

d. Details are needed on the database searched (species, variants allowed), search parameters (missed cleavages, modifications, FDR control etc.), and software used for the database search.

e. Similarly, details are needed on statistical methods, software, and specific GO/KEGG databases used for functional enrichment analyses.

9. Section 4.9 is missing critical methodological information for the transcriptomic analysis:

a. The software package/platform used should be specified for quality control of raw reads.

b. The mapping software/algorithm and reference genome used should be reported.

c. For hierarchical clustering, the distance measure (e.g. Euclidean) and linkage method (e.g. average) applied need to be stated.

d. Line 474 refers to defining "important enrichments" but a statistical cut-off or criteria is not provided. Please define how "importance" was determined.

e. Details of statistical analyses (packages, tests, FDR control etc.) are needed for differential expression, clustering and enrichment.

I hope these comments are helpful for improving clarity and completeness. Please let me know if any part needs more explanation.

Author Response

(The authors gave the same response as above.)

Round 2

Reviewer 1 Report

Comments and Suggestions for Authors

It is not clear to me where in the manuscript comments have been addressed.

I would expect the response to Comment 1 to be a description of your animals under Materials and Methods 4.1  It seems that the response to this comment was to add more background information in the introduction??

Comments on the Quality of English Language

There is a lot of new text in the introduction and it is unclear to me which comments it is intended to address. Can they revise their responses to make clear where in the manuscript they have addressed each comment?

Author Response

We are very sorry for the inconvenience caused by not clearly marking the response. We have re-uploaded the revised manuscript and marked it point-to-point

Reviewer 3 Report

Comments and Suggestions for Authors

The authors updated the manuscript according to previous suggestions. The quality of graphs is significantly improved. The authors also addressed methodological issues. The only comment refers now to the formatting but this could be corrected during the editing process. English editing would also be applied. Those minor issues could be easily corrected after paper acceptance.

Comments on the Quality of English Language

English editing would also be applied.

Author Response

Thank you very much for the positive comments.

We have sought English language editing services at the International Journal of Molecular Sciences.

Reviewer 4 Report

Comments and Suggestions for Authors

Dear Author(s),

I regret to inform you that the manuscript you have submitted lacks several critical methodological details regarding the proteomic sample preparation and data analysis. The omission of these crucial details raises significant concerns about the reproducibility and credibility of your work. Specifically:

1. Line 523, the sample contains 1% SDS, yet you have failed to mention how the SDS concentration was reduced before tryptic digestion, which is an absolute necessity to prevent enzyme denaturation.

2. The tryptic digestion protocol is entirely missing, including the duration, temperature, and enzyme-to-substrate ratio, which are essential parameters for this critical step.

3. In line 525, you state "Next, add 41µl TMT reagent to the sample," but you have neglected to provide the conditions (volume and buffer) of the sample, rendering this step impossible to replicate.

4. Regarding lines 529-530, you have described a protocol similar to a standard high pH fractionation method, but some crucial information is missing. Firstly, the mentioned buffer system may not achieve the desired high pH fractionation. Secondly, the LC gradient profile is not specified. Furthermore, you have failed to clarify whether the fractionation was UV-triggered or time-based, leaving this crucial aspect ambiguous.

5. Astonishingly, you have provided no methodological details whatsoever regarding the nLC-MS analysis, which is a fundamental component of your study.

6. Lastly, the manuscript lacks information on how the false discovery rate (FDR) for spectrum matching was determined, whether using Percolator, a fixed value FDR, or the target-decoy method.

The pervasive absence of these essential methodological details reflects a profound lack of scientific rigor and attention to detail. It is unacceptable for a manuscript of this caliber to omit such critical information, as it renders the entire work unreproducible and unverifiable.

Consequently, I must reject this manuscript in its current form. The authors have demonstrated a clear inability to present their proteomic work with the required level of detail and transparency expected in scientific publishing. Unless these glaring omissions are addressed comprehensively, I cannot recommend this manuscript for publication.

Author Response

Thank you very much for your professional advice.We are very sorry that we did not describe the details of the omics method in detail. We have added them in detail in the manuscript. We wish that you could give us a chance to reconsider our manuscript again.

Round 3

Reviewer 1 Report

Comments and Suggestions for Authors

I accept the revised manuscript

Comments on the Quality of English Language

I accept the revised manuscript

Author Response

Response:We sincerely thank you for your valuable comments that we have used to improve the quality of our manuscript and thank you very much for your recommendation.

Reviewer 4 Report

Comments and Suggestions for Authors

The methodology still requires significant clarification and revision before the work can be adequately assessed.

Line 550: Incorrectly states iodoacetamide was used - instead it should clearly indicate dithiothreitol as the standard reducing agent.

Lines 557: The full term for the abbreviation TEAB should be provided, and the pH of the solution should also be specified.

Line 562: The specific TMT kit used for protein modification should be mentioned, as different kits can result in different mass shifts (e.g., +229.16 or +304.2 Da), which is essential for accurate data interpretation and validation.

Line 577: The term "tryptophan peptide" requires clarification or elaboration.

Line 585: The description of the pre-column loading step is poorly written and incomprehensible, lacking essential details such as the pre-column loading time. Also, typically the flow rate should be at the dimension of µL/min for pre-column loading. Please be reminded that the flow rate of separation gradient is different from that of pre-column loading.

Line 601: Relegating the database matching parameters to supplementary material is unacceptable, as these are critical details that belong in the main text, presented in a clear and readable manner.

Line 610 states “We then used a hypergeometric distribution test to determine the significance of function enrichment in the differential protein list”, however Line 613 mentions “The GO function enrichment analysis of the DEPs was carried out using Blast 2 GO software.” Given that Blast2GO does not utilize a hypergeometric distribution test for analysis, this appears to be an inconsistency in the methodology as presented. It is important to clarify whether both approaches were applied or if there has been an error in reporting the statistical analysis method(s) used. Proper citation of the Blast2GO software, as well as the FatiGO tool referenced regarding multiple testing adjustment, should also be provided so readers can properly evaluate the bioinformatics approach taken.

Line 614 mischaracterizes the proteomics analysis method as "sequencing" rather than spectrum matching.

Regarding the newly added section:

Line 347: The statement "upregulated DEPs and DEGs were involved in the negative regulation of the apoptotic process" may require re-evaluation or rephrasing, as TP63, one of the upregulated DEPs, is actually positively related to apoptosis. Additionally, Eya3, one of the downregulated DEPs, is associated with GO term "GO:2001240 negative regulation of extrinsic apoptotic signaling pathway in absence of ligand," which further weakens the possibility of the stated hypothesis. By not acknowledging proteins that contradict the proposed role in negative apoptosis regulation, the conclusion drawn in line 347 is imprecise and not fully supported by the listed differential expression results.

In summary, the methodology is inadequately reported and inconsistent, while the interpretation of results extends beyond what is supported. This raises significant doubts about the authors' understanding of their own work. Major revisions could not fully remedy such a lack of comprehension. For these reasons, I cannot recommend accepting the manuscript and suggest rejection.

Author Response

We feel great thanks for your professional review work on our article.According to your nice suggestions, we have made extensive corrections to our manuscript.

Reviewer4#Round 3

The methodology still requires significant clarification and revision before the work can be adequately assessed.

Comments 1:Line 550: Incorrectly states iodoacetamide was used - instead it should clearly indicate dithiothreitol as the standard reducing agent.

Response:Thank you very much for your professional advice.We apologize for the error description.We have corrected in revised manuscript.

Comments 2:Lines 557: The full term for the abbreviation TEAB should be provided, and the pH of the solution should also be specified.

Response:Thank you very much for your professional advice.We apologize for the ignored of description.We have added in revised manuscript.The details are as follows:TEAB(Tetraethylammonium bromide,PH=6.14)

Comments 3: Line 562: The specific TMT kit used for protein modification should be mentioned, as different kits can result in different mass shifts (e.g., +229.16 or +304.2 Da), which is essential for accurate data interpretation and validation.

Response:Thank you very much for your professional advice.We apologize for the ignored of description.We have added in revised manuscript.The details are as follows:TMT reagent (TMTsixplex™ 6-plex,Thermo Fisher, USA)

Comments 4: Line 577: The term "tryptophan peptide" requires clarification or elaboration.

Response:Thank you very much for your professional advice.We apologize for the error description.We mistakenly wrote "Tryptic peptides" as "tryptophan peptide".We have corrected in revised manuscript.

Comments 5:Line 585: The description of the pre-column loading step is poorly written and incomprehensible, lacking essential details such as the pre-column loading time. Also, typically the flow rate should be at the dimension of µL/min for pre-column loading. Please be reminded that the flow rate of separation gradient is different from that of pre-column loading.

Response:Thank you very much for your professional advice.We apologize for the ignored of description.We have added in revised manuscript.

Comments 6:Line 601: Relegating the database matching parameters to supplementary material is unacceptable, as these are critical details that belong in the main text, presented in a clear and readable manner.

Response:Thank you very much for your professional advice.We apologize for the error description.We have corrected in revised manuscript,we changed the supplementary material of database matching parameters to Table2.

Comments 7:Line 610 states “We then used a hypergeometric distribution test to determine the significance of function enrichment in the differential protein list”, however Line 613 mentions “The GO function enrichment analysis of the DEPs was carried out using Blast 2 GO software.” Given that Blast2GO does not utilize a hypergeometric distribution test for analysis, this appears to be an inconsistency in the methodology as presented. It is important to clarify whether both approaches were applied or if there has been an error in reporting the statistical analysis method(s) used. Proper citation of the Blast2GO software, as well as the FatiGO tool referenced regarding multiple testing adjustment, should also be provided so readers can properly evaluate the bioinformatics approach taken.

Response:Thank you very much for your professional advice.We apologize for the error description.We use Hypergeometric distribution test and do not involve Blast 2 GO software.We have corrected and removed the incorrect description in the manuscript.

Comments 8:Line 614 mischaracterizes the proteomics analysis method as "sequencing" rather than spectrum matching.

Response:Thank you very much for your professional advice.We apologize for the error description.We have corrected in revised manuscript.

Regarding the newly added section:

Comments 9:Line 347: The statement "upregulated DEPs and DEGs were involved in the negative regulation of the apoptotic process" may require re-evaluation or rephrasing, as TP63, one of the upregulated DEPs, is actually positively related to apoptosis. Additionally, Eya3, one of the downregulated DEPs, is associated with GO term "GO:2001240 negative regulation of extrinsic apoptotic signaling pathway in absence of ligand," which further weakens the possibility of the stated hypothesis. By not acknowledging proteins that contradict the proposed role in negative apoptosis regulation, the conclusion drawn in line 347 is imprecise and not fully supported by the listed differential expression results.

Response:Thank you very much for your professional advice.We apologize for the incorrect description and expression.We have corrected in revised manuscript.